# Pregnancy complications recur independently of maternal vascular malperfusion lesions

**Julian K. Christians**[1,2,3,4]*, **Maria F. Huicochea Munoz**[5]

**1** Department of Biological Sciences, Simon Fraser University, Burnaby, Canada, **2** Centre for Cell Biology, Development and Disease, Simon Fraser University, Burnaby, Canada, **3** British Columbia Children's Hospital Research Institute, Vancouver, BC, Canada, **4** Women's Health Research Institute, Vancouver, BC, Canada, **5** Faculty of Health Sciences, McMaster University, Hamilton, Canada

* julian_christians@sfu.ca

**Data Availability Statement:** Data are publicly available (https://catalog.archives.gov/id/606622).

**Funding:** The authors received no specific funding for this work.

## Abstract

### Background

Spontaneous abortions, intrauterine growth restriction, and preeclampsia are thought to be caused by defective placentation and are associated with increased risk of adverse outcomes in subsequent pregnancies. However, it is not known whether the recurrence of adverse outcomes is associated with the recurrence of placental pathology. We hypothesized that recurrent maternal vascular malperfusion (MVM) underlies the recurrence of adverse outcomes.

### Methods

Using data from the National Collaborative Perinatal Project, we assessed the recurrence of pregnancy complications and MVM lesions (N = 3865), associations between a history of spontaneous abortions and MVM lesions or adverse outcomes in subsequent pregnancies (N = 8312), and whether the recurrence of pregnancy complications occurred independently of the presence of MVM lesions.

### Results

The odds of an MVM lesion were higher for a woman who had had an MVM lesion in a previous pregnancy (aOR = 1.6; 95% CI 1.3–1.9), although this was marginally non-significant after adjusting for covariates such as gestational age, race and BMI. The odds of preeclampsia, a small-for-gestational-age infant, premature delivery and early pregnancy loss were 2.7–5.0 times higher if there had been that same adverse outcome in a previous pregnancy. A history of spontaneous abortions was associated with higher risk of a small-for-gestational-age baby (aOR = 2.4; 95% CI 1.7–3.4) and prematurity (aOR = 5.1; 95% CI 2.3–11.5 for extremely preterm), but not preeclampsia. The recurrence of adverse outcomes was significant when restricting analyses to women without MVM lesions. Similarly, associations between adverse outcomes and previous spontaneous abortions were significant when statistically controlling for the presence of MVM lesions, or excluding pregnancies with MVM lesions.

**Competing interests:** The authors have declared that no competing interests exist.

## Conclusions

Women with adverse outcomes in one pregnancy are at higher risk of complications in subsequent pregnancies. However, there is significant recurrence of adverse outcomes even in the absence of MVM.

## Introduction

Placental dysfunction is thought to underlie diverse adverse pregnancy outcomes, including spontaneous abortions, intrauterine growth restriction, and preeclampsia [1], with spontaneous abortions and later pregnancy complications caused by different degrees of impaired placentation [2]. Consistent with this hypothesis, some circulating markers have been associated with both pregnancy loss and preeclampsia (e.g., PAPP-A [3–6]; sFlt-1, PlGF [7,8]). Furthermore, genetic polymorphisms in certain genes have been associated with both conditions (e.g., VEGF [9–12]; PAPP-A [13,14]; TNF-α [15,16]), potentially suggesting a common genetic susceptibility. Endothelial dysfunction has been proposed as a mechanism that might underlie these shared risks [17].

Given the potential shared etiology between preeclampsia and pregnancy loss, many studies have examined associations between adverse pregnancy outcomes in index pregnancies and spontaneous abortions in previous or subsequent pregnancies. Some find associations between the occurrence of spontaneous abortions and preeclampsia [2,18–23], low birthweight [2,18,20,23–26] and prematurity [2,18,23–27], although not all of these outcomes are consistently observed [28,29].

These studies have not examined the placental pathology that may underlie these associations, but impaired placentation is thought to result in maternal vascular malperfusion (MVM) lesions [30], which are associated with pregnancy complications [31–35]. Indeed, there are associations between MVM lesions in one pregnancy and the incidence of adverse outcomes in a subsequent pregnancy [36,37]. However, the association between MVM in one pregnancy and spontaneous abortion in another has not been investigated. Furthermore, it has not been shown that the recurrence of adverse outcomes in different pregnancies is associated with recurrent MVM pathology.

We hypothesized that certain factors predispose women to recurrent MVM, which increases the risk of spontaneous abortions, preeclampsia, intrauterine growth restriction, and prematurity, and that this mechanism underlies the previously observed associations between spontaneous abortions in one pregnancy and adverse outcomes in another. We therefore predicted that (1) MVM lesions or pregnancy complications in one pregnancy will be associated with an increased risk of MVM lesions or pregnancy complications in the subsequent pregnancy, (2) previous spontaneous abortions will be associated with an increased risk of MVM lesions, (3) previous spontaneous abortions will be associated with an increased risk of adverse outcomes, and (4) pregnancies with MVM lesions will have a higher incidence of adverse outcomes. Finally, if associations between spontaneous abortions and adverse outcomes are due to recurrent placental pathologies, we predict that (5) previous spontaneous abortions will not be associated with an increased risk of adverse outcomes independently of the presence of MVM lesions. We tested our predictions using data from the National Collaborative Perinatal Project.

## Materials and methods

The National Collaborative Perinatal Project (NCPP) has been described elsewhere [38], and its data are publicly available (https://catalog.archives.gov/id/606622). As we have described

previously [39], in over 90% of pregnancies, maternal race was categorized as white or black, and so analyses were restricted to these two races. We used only singleton pregnancies where offspring sex was assigned male or female; fetal and neonatal deaths were included, and cases were not excluded on the basis of maternal health conditions or congenital abnormalities. We excluded pregnancies where the gestational age was over 43 weeks; gestational age was calculated based on the last menstrual period to the nearest week. Within each gestational age category (described below), birthweights and placental weights were corrected for maternal race, offspring sex and gestational age using a general linear model, after first removing the top and bottom 0.5% of raw birthweights and placenta weights within each gestational age category to objectively exclude biologically implausible values [40]. Pregnancies ending before 24 weeks were excluded from all analyses except those of spontaneous abortions.

## Outcomes

To assess prematurity, we used a lower limit for gestational age based on the limit of viability [41], but otherwise used World Health Organization categories, i.e., extremely preterm (24 to 27 weeks, inclusive), very preterm (28 to 32 weeks, inclusive), moderate to late preterm (32 to 37 weeks, inclusive) and term (38 to 43 weeks, inclusive). In addition to prematurity, outcomes included spontaneous abortion (i.e., gestational age less than 20 weeks), small for gestational age (SGA, i.e., corrected birthweight below the $10^{th}$ percentile), preeclampsia (yes/no), survival (categorized as fetal death, death between birth and 120 days of age, or survival past 120 days), Apgar score at 1 and 5 minutes (categorized as 0–3, 4–6, or 7–10, where larger numbers are better). For preeclampsia, the rare cases of eclampsia were categorized as "yes", whereas "mild" preeclampsia was categorized as "no" because MVM lesions are more strongly associated with severe preeclampsia [30]. Preeclampsia was categorized as severe if one or more of the following symptoms was present: systolic blood pressure of 160 mmHg or higher, diastolic blood pressure of 110 mmHg or higher (on at least two occasions at least six hours apart), proteinuria of 5 grams or more, oliguria (400 cc or less in 24 hours), cerebral or visual disturbances, or pulmonary edema or cyanosis.

## MVM lesions

The NCPP data were collected > 50 years ago, and a comparison of variables available in the NCPP with lesions currently used to define MVM is provided in Table 1. MVM lesions included the presence of decidua vessel thrombosis, fibrinoid or atheroma, excessive fibrin deposition in the cytotrophoblast, at least one infarct measuring 3 cm or more, presence of hemorrhage, and syncytium-nuclear clumping that is excessive for term in a term placenta, or normal for term in a preterm placenta. While accelerated villous maturation is often included as an MVM lesion [42], it is difficult to diagnose in term placentas [39], and so syncytial clumping was used instead. We used two different approaches to establish evidence of MVM: (A) corrected placenta weight below the 10th percentile and one of the MVM lesions described above [33,43] (which we term "MVM$_{narrow}$"), or (B) one of the MVM lesions described, irrespective of placental weight [31,35] (which we term "MVM$_{broad}$"). Although not all of the lesions currently used to define MVM were available in the NCPP data, there is substantial variability in the criteria used among recent studies (Table 1), such that our study is as similar to some recent studies as the recent studies are to each other. We did not assess MVM for placentas with gestational age below 24 weeks, and so no analyses examined the association between spontaneous abortion and MVM lesions. Placental data were collected at 12 institutions. To reduce potential bias due to inconsistent collection of data, we excluded all data from

**Table 1. Comparison of lesions currently used to define MVM and variables available in NCPP data.**

| Current MVM criteria | NCPP variables |
|---|---|
| decidual vasculopathy[31,35,37,44,45], decidual vascular thrombosis[31]; decidual arteriopathy[30,42]; arterial thrombosis[42] | decidua vessel thrombosis |
| fibrinoid changes/necrosis in decidual vessels[31]; fibrinoid necrosis[42] | decidua vessel fibrinoid |
| acute atherosis[31,42], decidual vasculopathy atherosis [43]; | decidua vessel atheroma |
| increased perivillous fibrin[31]; increased perivillous fibrin deposition[35,37]; increased intervillous fibrin deposition[35,37]; fibrinoid deposition[44] | excessive fibrin deposition in the cytotrophoblast (as defined in NCPP documentation, "this refers to the fibrin or fibrinoid deposits which occur in cytotrophoblast cells as term is approached") |
| infarction[31,42]; villous infarction[35,37]; infarcts[44]; central (> 1 cm), full-thickness or multi-focal infarction [43]; placental infarction[30]; placental infarcts[45]; macroscopic infarction >5% of the placental parenchyma [46]; multifocal infarction[33] | at least one infarct measuring 3 cm or more |
| hypermature chorionic villi[31]; accelerated villous maturation[30,31,42–46]; advanced villous maturation [35,37]; increased syncytial knot prevalence[42]; abnormal placental villi[33] | syncytium-nuclear clumping that is excessive for term in a term placenta, or normal for term in a preterm placenta |
| low placental weight (placental weight<10th percentile) [33,42–44,46]; placental hypoplasia[30,42] | corrected placenta weight below the 10th percentile |
| retroplacental hematoma[31]; retroplacental hemorrhage [30,33,42] | presence of hemorrhage on maternal surface or depressed area caused by hemorrhage |
| distal villous hypoplasia[30,42,46]; mural hypertrophy [42]; chronic perivasculitis[42]; absence of spiral artery remodeling [42]; persistence of intramural endovascular trophoblast in the third trimester [42]; increased circulating nucleated red blood cells (NRBC)[46]; ischemia and shock villi [46]; eosinophilic necrosis of decidual arterial walls[43] | not described |

3 institutions where placental pathology data were available for less than 75% percent of pregnancies.

## Covariates

Gestational age, maternal age, maternal race, maternal body mass index (BMI), smoking (yes/ no) and interpregnancy interval were included as covariates. Maternal BMI was categorized as underweight (< 18.5), normal (18.5–25), overweight (>25–30) or obese (> 30). Interpregnancy interval was defined as the period between the end of the previous pregnancy and the last menstrual period before the current pregnancy, and was categorized as short (less than 18 months), medium (18 to 59 months, inclusive) or long (greater than 59 months), based on previously-reported associations with adverse outcomes [47,48].

## Recurrence of outcomes and pathology

To examine the recurrence of outcomes and pathology, we included only women with more than one singleton pregnancy in the dataset and compared the second eligible pregnancy in the dataset with the first. These criteria yielded 5889 eligible women. Excluding 440 women for whom the gestational age of the first and/ or second pregnancy was below 24 weeks, for which we did not assess MVM, there were 5449 women; assessment of MVM was available for both pregnancies for 3865 of these women. Comparison of characteristics of pregnancies with and

without MVM data is provided in the Table in S1 Table, including both women with and without more than one singleton pregnancy (study population described below). In the recurrence dataset, only 18 index pregnancies had an interpregnancy interval greater than 59 months and so these pregnancies were excluded from analyses involving interpregnancy interval.

## Associations between previous spontaneous abortions, MVM lesions, and outcomes

To examine associations between previous spontaneous abortions and outcomes, we compared women with 3 or more pregnancy losses at less than 20 weeks, with women with no prior losses (but at least one prior pregnancy), no record of sterility or infertility whose current pregnancy was conceived in 6 months or less. In both groups, we excluded women with more than 3 prior livebirths to ensure that the previous loss group included only women for whom at least half of previous pregnancies were losses, and not women who had had multiple losses simply as a result of numerous pregnancies. Where a woman had more than one pregnancy included in the NCCP, we included only her first study pregnancy. These criteria yielded 18077 eligible pregnancies. Excluding 3638 pregnancies with gestational age below 24 weeks (for which we did not assess MVM), there were 14439 pregnancies, of which assessment of MVM was available for 11753; comparison of characteristics of pregnancies with and without missing placental data is provided in the Table in S1 Table. For analyses including previous losses, 8312 pregnancies were available; 3199 pregnancies with 1 or 2 prior losses were excluded because we compared women with 3 or more pregnancy losses with no prior losses.

## Statistical analyses

Birthweight and placental weight were corrected for maternal race, offspring sex and gestational age using a general linear model (proc GLM, SAS, Version 9.4). Pregnancy outcomes and the presence of MVM lesions were analysed using logistic regression (proc LOGISTIC, SAS, Version 9.4).

## Results

### Prediction 1: MVM lesions or pregnancy complications in one pregnancy will be associated with an increased risk of MVM lesions or pregnancy complications in the subsequent pregnancy

Among women with more than one singleton pregnancy in the dataset, $MVM_{broad}$ lesions in one pregnancy were associated with increased odds of $MVM_{broad}$ lesions in the subsequent pregnancy (aOR = 1.6), although after adjusting for covariates such as gestational age, race and BMI, this was marginally non-significant (aOR = 1.2, P = 0.06, Table 2). Although the aOR for $MVM_{narrow}$ lesions was similar to that for $MVM_{broad}$ lesions, the recurrence was not significant, with or without adjustment for covariates, likely due to their very low frequency (Table 2). Preeclampsia, SGA, prematurity, spontaneous abortion, and poor Apgar scores in one pregnancy were associated with increased odds of the same adverse outcome occurring in the subsequent pregnancy, even after adjusting for interpregnancy interval, gestational age, maternal race, maternal BMI and smoking (Table 2). Adjusted odds ratios for the recurrence of adverse outcomes were highest for preeclampsia, SGA and prematurity (5.0, 4.6 and 3.7 respectively) and lowest for Apgar scores at 1 minute (1.4). Fetal and neonatal death did not show significant recurrence after correction for covariates, of which gestational age and race were significant (Table 2).

**Table 2. Recurrence of MVM and pregnancy outcomes between pregnancies.**

| | Occurrence in index pregnancy | Occurrence in previous pregnancy[1] | | Odds ratio[2] | P-value | Adjusted odds ratio[3] | P-value | Other significant terms in model[4] |
|---|---|---|---|---|---|---|---|---|
| | | Yes # (%) | No # (%) | (95% CI) | | (95% CI) | | |
| MVM$_{narrow}$ | Yes | 2 (3.2) | 60 (96.8) | 1.5 (0.4–6.1) | 0.59 | 1.4 (0.3–5.9) | 0.65 | Gestational age (P < 0.0001); Race (P = 0.006) |
| | No | 84 (2.2) | 3719 (97.8) | | | | | |
| MVM$_{broad}$ | Yes | 261 (29.8) | 616 (70.2) | 1.6 (1.3–1.9) | 0.0001 | 1.2 (1.0–1.5) | 0.06 | Gestational age (P < 0.0001); Race (P = 0.02); BMI (P = 0.02) |
| | No | 655 (21.1) | 2446 (78.9) | | | | | |
| Preeclampsia | Yes | 18 (19.8) | 73 (80.2) | 6.5 (3.8–11.2) | 0.0001 | 5.0 (2.9–8.7) | 0.0001 | Race (P < 0.0001); BMI (P < 0.002) |
| | No | 170 (3.6) | 4498 (96.4) | | | | | |
| SGA | Yes | 145 (31.1) | 322 (68.9) | 5.2 (4.2–6.5) | 0.0001 | 4.6 (3.7–5.8) | 0.0001 | Smoking (P < 0.0001) |
| | No | 385 (8.0) | 4440 (92.0) | | | | | |
| Prematurity | Extreme | 32 (40.5) | 47 (59.5) | 4.3 (2.7–6.8) | 0.0001 | 3.7 (2.3–5.9) | 0.0001 | Race (P < 0.0001); BMI (P = 0.0004) |
| | Very | 91 (43.3) | 119 (56.7) | 4.8 (3.6–6.4) | | 3.8 (2.8–5.1) | | |
| | Moderate | 369 (40.7) | 537 (59.3) | 4.3 (3.7–5.1) | | 3.7 (3.1–4.3) | | |
| | Term | 583 (13.7) | 3671 (86.3) | | | | | |
| Spontaneous abortion before 20 weeks | Yes | 15 (9.2) | 149 (90.9) | 3.0 (1.7–5.3) | 0.0001 | 2.7 (1.5–5.1) | 0.001 | Race (P = 0.04); BMI (P = 0.02) |
| | No | 184 (3.2) | 5541 (96.8) | | | | | |
| Survival | Fetal death | 11 (12.6) | 76 (87.4) | 3.8 (2.0–7.2) | 0.0001 | 2.3 (1.0–5.0) | 0.11 | Gestational age (P < 0.0001); Race (P = 0.01) |
| | Death before 120 days | 10 (8.0) | 115 (92.0) | 2.3 (1.2–4.4) | | 1.5 (0.7–3.0) | | |
| | Survival past 120 days | 193 (3.7) | 5044 (96.3) | | | | | |
| Apgar score at 1 minute | 0–3 | 55 (28.4) | 139 (71.6) | 1.6 (1.2–2.2) | 0.0002 | 1.5 (1.1–2.1) | 0.0006 | Gestational age (P < 0.0001) |
| | 4–6 | 147 (25.7) | 424 (74.3) | 1.4 (1.1–1.7) | | 1.4 (1.2–1.7) | | |
| | 7–10 | 747 (19.8) | 3028 (80.2) | | | | | |
| Apgar score at 5 minutes | 0–3 | 9 (14.8) | 52 (85.3) | 3.9 (1.9–8.0) | 0.0002 | 3.1 (1.4–6.6) | 0.006 | Gestational age (P < 0.0001) |
| | 4–6 | 9 (8.3) | 99 (91.7) | 2.0 (1.0–4.1) | | 1.8 (0.9–3.6) | | |
| | 7–10 | 195 (4.3) | 4350 (95.7) | | | | | |

[1] For outcomes with multiple levels of adverse outcomes, all levels of adverse outcome (e.g., extreme, very and moderate preterm) were combined for the previous pregnancy.

[2] All odds ratios were calculated as the odds of the adverse outcome relative to the odds of the best outcome where there were multiple levels of adverse outcomes.

[3] Adjusted odds ratios are from logistic regression including interpregnancy interval, gestational age, maternal race, maternal BMI and smoking. Analyses of prematurity and spontaneous abortion before 20 weeks did not include gestational age.

[4] Adjusted odds ratios for other terms are provided in the Table in S3 Table.

To assess whether the recurrence of pregnancy complications occurred in the absence of recurrent MVM lesions, we restricted analyses to women who showed no MVM lesion in either pregnancy. Preeclampsia, SGA and prematurity in one pregnancy were associated with increased odds of the same adverse outcome occurring in the subsequent pregnancy after adjusting for covariates (Table in S2 Table). Moreover, the adjusted odds ratios were similar whether women with MVM lesions were excluded (Table in S2 Table) or not (Table 2) for the recurrence of preeclampsia (excluding: 4.9–5.2 vs. not excluding: 5.0), SGA (4.1–4.6 vs. 4.6) and prematurity (3.3–8.7 vs. 3.7). Similar results were observed whether using our "narrow" or "broad" criteria for MVM lesions, although the recurrence of Apgar scores was not statistically significant when excluding pregnancies showing $MVM_{broad}$ lesions (Table in S2 Table). Given the broader inclusion criteria, $MVM_{broad}$ lesions were frequent (Table 2) and so their exclusion reduced sample sizes substantially, reducing statistical power.

## Prediction 2: Previous spontaneous abortions will be associated with an increased risk of MVM lesions in index pregnancy

We compared women with 3 or more pregnancy losses at less than 20 weeks with women with no prior losses. Previous spontaneous abortions were associated with increased odds of $MVM_{narrow}$ lesions (aOR = 2.2), but not $MVM_{broad}$ lesions (aOR = 0.9), in the index pregnancy when adjusting for interpregnancy interval, gestational age, maternal age, maternal race, maternal BMI and smoking (Table 3).

## Prediction 3: Previous spontaneous abortions will be associated with an increased risk of adverse outcomes in index pregnancy

Adjusting for covariates, previous spontaneous abortions were associated with increased odds of SGA (aOR = 2.4), extremely (aOR = 5.1) or very preterm birth (aOR = 2.2), spontaneous abortion before 20 weeks (aOR = 2.1), fetal death at 24 weeks or later (aOR = 4.6), and low Apgar scores at 1 (aOR = 1.8) and 5 minutes (aOR = 4.5; Table 3). Previous spontaneous abortions were not associated with increased odds of preeclampsia (aOR = 1.1; Table 3).

## Prediction 4: Pregnancies with MVM lesions will have a higher incidence of adverse outcomes

The presence of MVM, defined using our "narrow" criteria, was associated with increased odds of preeclampsia (aOR = 2.6), SGA (aOR = 11.5), prematurity (aOR = 3.9–4.7), and fetal death (aOR = 12.5), before and after adjustment for covariates (Table 4). Using our "broad" criteria, MVM lesions were associated with higher odds of SGA (aOR = 1.2) and prematurity (aOR = 9.8–24.7), and fetal death (aOR = 2.1) but not preeclampsia or Apgar scores after adjustment for covariates (Table 5).

## Prediction 5: Previous spontaneous abortions will not be associated with an increased risk of adverse outcomes independently of the presence of MVM lesions

Adjusting for covariates including the presence of MVM lesions, previous spontaneous abortions were associated with increased odds of SGA, prematurity and fetal death at 24 weeks or later, and lower Apgar scores at 1 and 5 minutes (Table 6). Results were similar whether using our narrow or broad criteria to define the presence of MVM lesions. Adjusted odds ratios for the association with previous spontaneous abortions were similar whether adjusting for the presence of MVM lesions (Table 6) or not (Table 3) for SGA (adjusting for MVM lesions: 1.9–

**Table 3. Associations between previous spontaneous abortions and pregnancy outcomes.**

| | Occurrence in index pregnancy | Previous spontaneous abortions | | Odds ratio[1] | P-value | Adjusted odds ratio[2] | P-value | Other significant terms in model[3] |
|---|---|---|---|---|---|---|---|---|
| | | Yes # (%) | No # (%) | (95% CI) | | (95% CI) | | |
| MVM$_{narrow}$ | Yes | 12 (7.0) | 160 (93.0) | 3.0 (1.6–5.5) | 0.0003 | 2.2 (1.2–4.4) | 0.02 | Interpregnancy interval (P = 0.04); Gestational age (P < 0.0001) |
| | No | 198 (2.4) | 7942 (97.6) | | | | | |
| MVM$_{broad}$ | Yes | 53 (2.7) | 1879 (97.3) | 1.1 (0.8–1.5) | 0.60 | 0.9 (0.6–1.3) | 0.51 | Gestational age (P < 0.0001) |
| | No | 165 (2.5) | 6368 (97.5) | | | | | |
| Preeclampsia | Yes | 8 (3.3) | 234 (96.7) | 1.4 (0.7–2.8) | 0.38 | 1.1 (0.5–2.4) | 0.72 | Age (P = 0.001); Gestational age (P = 0.03); Race (P < 0.0001); BMI (P < 0.0001); Smoking (P = 0.03) |
| | No | 224 (2.4) | 9056 (97.6) | | | | | |
| SGA | Yes | 50 (5.9) | 801 (94.1) | 2.8 (2.0–3.9) | 0.0001 | 2.4 (1.7–3.4) | 0.0001 | Age (P = 0.004); Gestational age (P = 0.008); BMI (P < 0.0001); Smoking (P < 0.0001) |
| | No | 199 (2.2) | 8983 (97.8) | | | | | |
| Prematurity | Extreme | 9 (8.7) | 95 (91.3) | 3.8 (1.9–7.5) | 0.0009 | 5.1 (2.3–11.5) | 0.0001 | Age (P = 0.02); Race (P < 0.0001); BMI (P < 0.0001); Smoking (P < 0.0001) |
| | Very | 13 (4.1) | 302 (95.9) | 1.7 (1.0–3.0) | | 2.2 (1.2–4.1) | | |
| | Moderate | 41 (2.6) | 1522 (97.4) | 1.1 (0.8–1.5) | | 1.3 (0.9–1.8) | | |
| | Term | 201 (2.5) | 7966 (97.5) | | | | | |
| Spontaneous abortion before 20 weeks | Yes | 21 (10.8) | 174 (89.2) | 4.4 (2.8–7.0) | 0.0001 | 2.1 (1.2–3.6) | 0.005 | Interpregnancy interval (P = 0.0002); Age (P < 0.0001); BMI (P = 0.03); Smoking (P = 0.05) |
| | No | 272 (2.7) | 9932 (97.3) | | | | | |
| Survival | Fetal death | 18 (13.3) | 117 (86.7) | 6.2 (3.7–10.3) | 0.0001 | 4.6 (2.3–9.0) | 0.0001 | Interpregnancy interval (P = 0.01); Gestational age (P < 0.0001); Race (P = 0.006) |
| | Death before 120 days | 7 (3.5) | 195 (96.5) | 1.4 (0.7–3.1) | | 1.3 (0.6–3.0) | | |
| | Survival past 120 days | 239 (2.4) | 9573 (97.6) | | | | | |
| Apgar score at 1 minute | 0–3 | 19 (4.5) | 405 (95.5) | 2.1 (1.3–3.4) | 0.001 | 1.8 (1.1–3.0) | 0.02 | Interpregnancy interval (P = 0.002); Age (P = 0.03); Gestational age (P < 0.0001); Race (P < 0.03) |
| | 4–6 | 39 (3.4) | 1110 (96.6) | 1.6 (1.1–2.2) | | 1.5 (1.0–2.1) | | |
| | 7–10 | 166 (2.2) | 7428 (97.8) | | | | | |
| Apgar score at 5 minutes | 0–3 | 9 (8.1) | 102 (91.9) | 3.6 (1.8–7.2) | 0.001 | 4.5 (2.0–10.1) | 0.0007 | Interpregnancy interval (P = 0.02); Gestational age (P < 0.0001) |
| | 4–6 | 4 (1.6) | 240 (98.4) | 0.7 (0.2–1.8) | | 0.6 (0.2–1.6) | | |
| | 7–10 | 216 (2.4) | 8754 (97.6) | | | | | |

[1] All odds ratios were calculated as the odds of the adverse outcome relative to the odds of the best outcome where there were multiple levels of adverse outcomes.

[2] Adjusted odds ratios are from logistic regression including interpregnancy interval, gestational age, maternal age, maternal race, maternal BMI and smoking. Analyses of prematurity and spontaneous abortion before 20 weeks did not include gestational age.

[3] Adjusted odds ratios for other terms are provided in the Table in S4 Table.

**Table 4. Associations between MVM$_{narrow}$ lesions and pregnancy outcomes.**

| | | MVM$_{narrow}$ | | Odds ratio[1] | P-value | Adjusted odds ratio[2] | P-value | Other significant terms in model |
|---|---|---|---|---|---|---|---|---|
| | Occurrence in pregnancy | Yes # (%) | No # (%) | (95% CI) | | (95% CI) | | |
| Preeclampsia | Yes | 14 (4.6) | 294 (95.4) | 2.4 (1.4–4.1) | 0.002 | 2.6 (1.5–4.6) | 0.0009 | Age (P < 0.0001); Race (P < 0.0001); BMI (P < 0.0001); Smoking (P = 0.01) |
| | No | 213 (2.0) | 10621 (98.0) | | | | | |
| SGA | Yes | 115 (10.9) | 943 (89.1) | 10.2 (7.9–13.3) | 0.0001 | 11.5 (8.7–15.2) | 0.0001 | Age (P = 0.0003); Gestational age (P < 0.0001); BMI (P < 0.0001); Smoking (P < 0.0001) |
| | No | 125 (1.2) | 10495 (98.8) | | | | | |
| Prematurity | Extreme | 6 (6.0) | 94 (94.0) | 4.5 (1.9–10.6) | 0.0001 | 4.7 (2.0–11.0) | 0.0001 | Age (P = 0.02); Race (P < 0.0001); BMI (P < 0.0001); Smoking (P < 0.0001) |
| | Very | 21 (6.2) | 318 (93.8) | 4.7 (2.9–7.6) | | 4.7 (2.9–7.8) | | |
| | Moderate | 89 (5.2) | 1625 (94.8) | 3.9 (3.0–5.1) | | 3.9 (2.9–5.2) | | |
| | Term | 133 (1.4) | 9467 (98.6) | | | | | |
| Survival | Fetal death | 34 (26.0) | 97 (74.0) | 19.2 (12.7–29.1) | 0.0001 | 12.5 (7.8–19.9) | 0.0001 | Interpregnancy interval (P = 0.05); Age (P = 0.001); Gestational age (P < 0.0001); Race (P = 0.008) |
| | Death before 120 days | 11 (4.7) | 225 (95.3) | 2.7 (1.4–5.0) | | 1.7 (0.9–3.3) | | |
| | Survival past 120 days | 204 (1.8) | 11182 (98.2) | | | | | |
| Apgar score at 1 minute | 0–3 | 20 (3.7) | 526 (96.3) | 2.2 (1.4–3.5) | 0.005 | 1.6 (1.0–2.7) | 0.10 | Interpregnancy interval (P = 0.002); Age (P = 0.008); Gestational age (P < 0.0001); BMI (P < 0.02) |
| | 4–6 | 24 (1.6) | 1436 (98.4) | 1.0 (0.6–1.5) | | 0.9 (0.6–1.4) | | |
| | 7–10 | 154 (1.7) | 8820 (98.3) | | | | | |
| Apgar score at 5 minutes | 0–3 | 6 (4.6) | 124 (95.4) | 2.8 (1.2–6.4) | 0.0002 | 1.6 (0.6–4.1) | 0.02 | Interpregnancy interval (P = 0.02); Gestational age (P < 0.0001) |
| | 4–6 | 13 (4.6) | 272 (95.4) | 2.8 (1.6–4.9) | | 2.2 (1.2–3.9) | | |
| | 7–10 | 183 (1.7) | 10556 (98.3) | | | | | |

[1] All odds ratios were calculated as the odds of the adverse outcome relative to the odds of the best outcome where there were multiple levels of adverse outcomes.

[2] Adjusted odds ratios are from logistic regression including interpregnancy interval, gestational age, maternal age, maternal race, maternal BMI and smoking. Analyses of prematurity did not include gestational age.

2.2 vs. not adjusting: 2.4), prematurity (2.9–3.1 vs. 2.2 for very premature birth), and fetal death (4.5–5.0 vs. 4.6). Previous spontaneous abortions were not associated with preeclampsia with (Table 6) or without (Table 3) inclusion of MVM lesions (defined using narrow or broad criteria) in the model.

Restricting analyses to pregnancies with no signs of MVM lesions, previous spontaneous abortions remained associated with increased odds of SGA, prematurity, and lower Apgar scores (Table 7). However, the association between previous spontaneous abortions and fetal death at 24 weeks or later was no longer significant when removing pregnancies with MVM lesions (Table 7). Similar results were observed whether using our "narrow" or "broad" criteria

**Table 5. Associations between MVM$_{broad}$ lesions and pregnancy outcomes.**

| | | MVM$_{broad}$ | | Odds ratio[1] | P-value | Adjusted odds ratio[2] | P-value | Other significant terms in model |
|---|---|---|---|---|---|---|---|---|
| | Occurrence in pregnancy | Yes # (%) | No # (%) | (95% CI) | | (95% CI) | | |
| Preeclampsia | Yes | 83 (26.3) | 232 (73.7) | 1.3 (1.0–1.6) | 0.09 | 1.2 (0.9–1.6) | 0.24 | Age (P < 0.0001); Race (P < 0.0001); BMI (P < 0.0001); Smoking (P = 0.03) |
| | No | 2455 (22.3) | 8576 (77.7) | | | | | |
| SGA | Yes | 264 (24.3) | 820 (75.7) | 1.1 (1.0–1.3) | 0.10 | 1.2 (1.0–1.4) | 0.02 | Interpregnancy interval (P = 0.03); Age (P = 0.0002); Gestational age (P < 0.0005); BMI (P < 0.0001); Smoking (P < 0.0001) |
| | No | 2388 (22.1) | 8399 (77.9) | | | | | |
| Prematurity | Extreme | 69 (68.3) | 32 (31.7) | 14.0 (9.2–21.4) | 0.0001 | 14.8 (9.6–22.8) | 0.0001 | Race (P < 0.0001); BMI (P < 0.0001); Smoking (P < 0.0001) |
| | Very | 272 (78.4) | 75 (21.6) | 23.6 (18.2 = 30.7) | | 24.7 (18.8–32.3) | | |
| | Moderate | 1042 (59.6) | 707 (40.4) | 9.6 (8.6–10.7) | | 9.8 (8.8–11.1) | | |
| | Term | 1301 (13.3) | 8473 (86.7) | | | | | |
| Survival | Fetal death | 90 (62.5) | 54 (37.5) | 6.1 (4.3–8.5) | 0.0001 | 2.1 (1.4–3.1) | 0.0001 | Age (P < 0.0001); Gestational age (P < 0.0001); Race (P = 0.003) |
| | Death before 120 days | 100 (41.0) | 144 (59.0) | 2.5 (1.9–3.3) | | 0.7 (0.5–0.9) | | |
| | Survival past 120 days | 2494 (21.5) | 9089 (78.5) | | | | | |
| Apgar score at 1 minute | 0–3 | 165 (29.6) | 392 (70.4) | 1.6 (1.3–1.9) | 0.0001 | 1.0 (0.8–1.2) | 0.53 | Interpregnancy interval (P = 0.001); Age (P = 0.007); Gestational age (P < 0.0001); BMI (P < 0.02) |
| | 4–6 | 349 (23.6) | 1132 (76.4) | 1.2 (1.0–1.3) | | 1.1 (0.9–1.3) | | |
| | 7–10 | 1918 (21.0) | 7214 (79.0) | | | | | |
| Apgar score at 5 minutes | 0–3 | 54 (40.6) | 79 (59.4) | 2.5 (1.8–3.6) | 0.0001 | 0.7 (0.4–1.0) | 0.15 | Interpregnancy interval (P = 0.02); Gestational age (P < 0.0001) |
| | 4–6 | 93 (31.9) | 199 (68.2) | 1.7 (1.3–2.2) | | 0.9 (0.7–1.2) | | |
| | 7–10 | 2332 (21.4) | 8586 (78.64) | | | | | |

[1] All odds ratios were calculated as the odds of the adverse outcome relative to the odds of the best outcome where there were multiple levels of adverse outcomes.

[2] Adjusted odds ratios are from logistic regression including interpregnancy interval, gestational age, maternal age, maternal race, maternal BMI and smoking. Analyses of prematurity did not include gestational age.

for MVM lesions (Table 7). Adjusted odds ratios were similar whether pregnancies with MVM lesions were removed (Table 7) or not (Table 3) for SGA (excluding pregnancies with MVM lesions: 1.8 vs. including: 2.4) and prematurity (2.7–3.8 vs. 2.2 for very premature birth).

## Discussion

MVM lesions are thought to reflect placental dysfunction that may underlie spontaneous abortions, preeclampsia and intrauterine growth restriction. We examined whether recurrence of pathology might underlie the recurrence of adverse outcomes. The odds of an MVM lesion were 1.6 times higher for a woman who had had an MVM lesion in a previous pregnancy,

**Table 6. Associations of previous spontaneous abortions and pregnancy outcomes including the presence of MVM lesions as a covariate.**

| | Including MVM$_{narrow}$ as a covariate | | | | Including MVM$_{broad}$ as a covariate | | | |
| | Effect of previous spontaneous abortions | | Effect of MVM | | Effect of previous spontaneous abortions | | Effect of MVM | |
| | aOR[1] | P-value | aOR[1] | P-value | aOR[1] | P-value | aOR[1] | P-value |
|---|---|---|---|---|---|---|---|---|
| Preeclampsia | 1.0 (0.4–2.4) | 0.98 | 3.5 (1.8–6.7) | 0.0002 | 1.2 (0.5–2.7) | 0.69 | 1.3 (0.9–1.9) | 0.14 |
| SGA | 1.9 (1.3–2.9) | 0.002 | 10.6 (7.5–15.0) | 0.0001 | 2.2 (1.5–3.2) | 0.0001 | 1.2 (1.0–1.5) | 0.04 |
| Prematurity | | 0.002 | | 0.0001 | | 0.0005 | | 0.0001 |
| Extreme | 4.1 (1.3–12.5) | | 3.1 (0.9–10.2) | | 5.7 (2.0–16.4) | | 14.0 (8.3–23.4) | |
| Very | 2.9 (1.5–5.6) | | 4.0 (2.2–7.2) | | 3.1 (1.5–6.3) | | 29.7 (21.4–41.1) | |
| Moderate | 1.3 (0.8–1.9) | | 3.5 (2.5–4.9) | | 1.3 (0.8–2.1) | | 9.9 (8.6–11.3) | |
| Survival | | 0.006 | | 0.0001 | | 0.0008 | | 0.008 |
| Fetal death | 4.5 (1.8–11.3) | | 15.4 (8.6–27.6) | | 5.0 (2.2–11.6) | | 2.0 (1.2–3.4) | |
| Death before 120 days | 1.1 (0.4–2.8) | | 1.4 (0.6–3.3) | | 1.1 (0.4–2.7) | | 0.8 (0.5–1.1) | |
| Apgar at 1 minute | | 0.03 | | 0.18 | | 0.02 | | 0.14 |
| 0–3 | 1.6 (0.9–2.9) | | 1.8 (0.9–3.2) | | 1.8 (1.0–3.1) | | 1.3 (1.0–1.7) | |
| 4–6 | 1.6 (1.1–2.4) | | 0.9 (0.5–1.6) | | 1.6 (1.1–2.3) | | 1.1 (0.9–1.3) | |
| Apgar at 5 minutes | | 0.0009 | | 0.08 | | 0.001 | | 0.30 |
| 0–3 | 5.1 (2.0–12.5) | | 2.2 (0.8–6.2) | | 5.0 (2.0–12.2) | | 0.7 (0.4–1.1) | |
| 4–6 | 0.5 (0.2–1.7) | | 2.0 (0.9–4.1) | | 0.5 (0.2–1.7) | | 0.9 (0.6–1.3) | |

[1] Adjusted odds ratios are from logistic regression including interpregnancy interval, gestational age, maternal age, maternal race, maternal BMI and smoking. Analyses of prematurity did not include gestational age.

using our broad definition of MVM lesions, although this was marginally non-significant after adjustment for covariates. The odds of an adverse outcome were 2.7–4.9 times higher if there

**Table 7. Associations of previous spontaneous abortions and pregnancy outcomes excluding cases with MVM lesions.**

| | Excluding MVM$_{narrow}$ lesions | | Excluding MVM$_{broad}$ lesions | |
| | aOR[1] | P-value | aOR[1] | P-value |
|---|---|---|---|---|
| Preeclampsia | 0.8 (0.3–2.1) | 0.71 | 1.0 (0.4–2.9) | 0.96 |
| SGA | 1.8 (1.1–2.8) | 0.01 | 1.8 (1.2–2.9) | 0.01 |
| Prematurity | | 0.003 | | 0.003 |
| Extreme | 4.7 (1.5–14.3) | | 12.0 (2.3–63.5) | |
| Very | 2.7 (1.3–5.4) | | 3.8 (1.2–12.0) | |
| Moderate | 1.3 (0.8–1.9) | | 1.3 (0.7–2.3) | |
| Survival | | 0.12 | | 0.54 |
| Fetal death | 2.9 (1.0–8.8) | | 1.7 (0.3–9.1) | |
| Death before 120 days | 0.7 (0.2–2.2) | | 0.5 (0.1–2.4) | |
| Apgar at 1 minute | | 0.03 | | 0.04 |
| 0–3 | 1.6 (0.9–3.0) | | 1.9 (1.0–3.8) | |
| 4–6 | 1.6 (1.1–2.4) | | 1.6 (1.0–2.4) | |
| Apgar at 5 minutes | | 0.03 | | 0.15 |
| 0–3 | 3.5 (1.3–9.5) | | 3.2 (0.9–12.0) | |
| 4–6 | 0.5 (0.2–1.8) | | 0.5 (0.1–2.3) | |

[1] Adjusted odds ratios are from logistic regression including interpregnancy interval, gestational age, maternal age, maternal race, maternal BMI and smoking. Analyses of prematurity did not include gestational age.

had been that adverse outcome in a previous pregnancy. As described by others [2,18,20,23–27], previous spontaneous abortions were associated with 2.4 times higher odds of SGA and 5.1 times higher odds of extreme prematurity. Furthermore, we also showed that previous spontaneous abortions were associated with increased risk of low Apgar scores. However, we did not find an association between previous spontaneous abortions and preeclampsia.

The presence of MVM lesions was associated with higher odds of preeclampsia, SGA, prematurity, and fetal death, as previously observed [30], as well as low Apgar scores. These observations, largely consistent with previous work, led us to hypothesize that the associations between previous spontaneous abortions and subsequent adverse outcomes, and the recurrence of the same adverse outcome, were due to the recurrence of MVM lesions. However, the associations between previous spontaneous abortions and adverse outcomes generally remained significant and similar in magnitude even when controlling for the presence MVM lesions, or when removing pregnancies with MVM lesions. Furthermore, among women with more than one pregnancy in the dataset, the recurrence of adverse outcomes was significant when restricting analyses to women without MVM lesions in either pregnancy. These results suggested that the recurrence of adverse outcomes, and the associations between spontaneous abortions and adverse outcomes in subsequent pregnancies, can occur independently of recurrent MVM lesions. While it is possible that recurrent pathology did account for some cases of recurrent adverse outcomes, the adjusted odds ratios for associations with previous abortions and for the recurrence of adverse outcomes were generally not reduced by controlling for the presence MVM lesions or removing pregnancies with MVM lesions, suggesting that a substantial component of recurrence risk is independent of pathology. Previous studies have shown that the presence of MVM in one pregnancy is associated with increased risk of adverse outcome in a subsequent pregnancy [36,37,49]. Our results indicate that the MVM lesion itself does not have to recur to result in an adverse outcome, and suggest that other, unmeasured confounding factors may be responsible for these associations.

We used two different approaches described in the literature to define the presence of MVM, differing only in whether low placental weight was required for diagnosis (MVM$_{narrow}$) [33,43] or not (MVM$_{broad}$) [31,35]. The two approaches generally yielded similar results, except that MVM$_{narrow}$ lesions were associated with higher odds of preeclampsia whereas MVM$_{broad}$ lesions were not. These results suggest that the "narrow" approach provided a more specific, informative diagnosis.

Approximately 2% of placentas showed MVM$_{narrow}$ lesions whereas 23% showed MVM$_{broad}$ lesions. Our prevalence of MVM$_{broad}$ lesions is lower than previous reports (30.5% [35]; 35.7% [50]; 39.9% [31]; 46.7–49.7% [44]), perhaps because healthy pregnancies were less likely to have been examined in some of these other studies. Our prevalence of MVM$_{narrow}$ lesions is similar to that in studies that used a similar definition (8.4% [33]; 0–4.2% including ethnicities included in the present study [43]).

We acknowledge that our study used data collected > 50 years ago, and that assessment of placental pathology has changed in that time. For example, the term distal villous hypoplasia was not used in the histological assessment of the placentas. Despite this, our observed rates of MVM$_{narrow}$ are in the range of those of more recent studies, as discussed above. A strength of the present study is that placental pathology was performed consistently and collected for healthy, uncomplicated pregnancies, with pathology data available for over 80% of pregnancies. Moreover, we reproduced the association between MVM lesions and preeclampsia [31,33,35]. Furthermore, the lesions used to define MVM are not consistent among current studies (Table 1), and some authors require lesions to be combined with low placental weight whereas others do not, resulting in very different rates of MVM. We investigated both approaches, and the criteria we used were very similar to some recent work. Finally, while the

assessment of pathology has evolved, the biology underlying the pathology and the associations between placental function and adverse outcomes are not expected to have changed.

## Conclusions

Women with spontaneous abortions or other adverse outcomes in a previous pregnancy are at higher risk of adverse outcomes in subsequent pregnancies. However, the recurrence of outcomes thought to be associated with MVM lesions occurs even in the absence of placental pathology. In many cases, the recurrence of an adverse outcomes may not be due to an intrinsic predisposition to a specific placental pathology, but rather may be caused by other aspects of maternal physiology beyond the placenta.

## Supporting information

**S1 Table. Characteristics of pregnancies with and without missing placental data, including only pregnancies at 24 weeks or later and excluding 3 institutions where placental pathology data were available for less than 75% percent of pregnancies.**
(DOCX)

**S2 Table. Recurrence of outcomes between pregnancies, restricting analyses to women not showing MVM lesions in either pregnancy.**
(DOCX)

**S3 Table. Adjusted odds ratios for all covariates in analyses of recurrence of MVM and pregnancy outcomes between pregnancies.**
(DOCX)

**S4 Table. Adjusted odds ratios for all covariates in analyses of effects of previous spontaneous abortions on pregnancy outcomes.**
(DOCX)

## Acknowledgments

We thank the U.S. National Archives for making the National Collaborative Perinatal Project data freely available, and David Grynspan and Jefferson Terry for helpful discussion.

## Author Contributions

**Conceptualization:** Julian K. Christians, Maria F. Huicochea Munoz.

**Formal analysis:** Julian K. Christians.

**Writing – original draft:** Julian K. Christians.

**Writing – review & editing:** Maria F. Huicochea Munoz.

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
