## [Decision Letter · Decision Letter 0]

20 Dec 2019

PONE-D-19-32724

Pregnancy complications recur independently of maternal vascular malperfusion lesions

PLOS ONE

Dear Christians,

Thank you for submitting your manuscript to PLOS ONE. After careful consideration, we feel that it has merit but does not fully meet PLOS ONE’s publication criteria as it currently stands. Therefore, we invite you to submit a revised version of the manuscript that addresses the points raised during the review process.

We would appreciate receiving your revised manuscript by Feb 03 2020 11:59PM. To enhance the reproducibility of your results, we recommend that if applicable you deposit your laboratory protocols in protocols.io, where a protocol can be assigned its own identifier (DOI) such that it can be cited independently in the future. For instructions see: http://journals.plos.org/plosone/s/submission-guidelines#loc-laboratory-protocols

We look forward to receiving your revised manuscript.

Kind regards,

Frank T. Spradley

Academic Editor

PLOS ONE

2. Please note that all PLOS journals ask authors to adhere to our policies for sharing of data and materials: https://journals.plos.org/plosone/s/data-availability. According to PLOS ONE’s Data Availability policy, we require that the minimal dataset underlying results reported in the submission must be made immediately and freely available at the time of publication. As such, please remove any instances of 'unpublished data' or 'data not shown' in your manuscript and replace these with either the relevant data (in the form of additional figures, tables or descriptive text, as appropriate), a citation to where the data can be found, or remove altogether any statements supported by data not presented in the manuscript.

3. We noticed you have some minor occurrence(s) of overlapping text with the following previous publication(s), which needs to be addressed:

https://doi.org/10.1016/j.placenta.2019.01.012

https://doi.org/10.1177%2F1093526619852871

In your revision ensure you cite all your sources (including your own works), and quote or rephrase any duplicated text outside the Methods section. Further consideration is dependent on these concerns being addressed.

Reviewers' comments:

Reviewer's Responses to Questions

**Comments to the Author**

1. Is the manuscript technically sound, and do the data support the conclusions?

Reviewer #1: Partly

2. Has the statistical analysis been performed appropriately and rigorously? 

Reviewer #1: Yes

3. Have the authors made all data underlying the findings in their manuscript fully available?

Reviewer #1: Yes

4. Is the manuscript presented in an intelligible fashion and written in standard English?

Reviewer #1: Yes

5. Review Comments to the Author

Reviewer #1: The authors take advantage of the large number of patients followed and the high quality placental pathology in the Collaborative Perinatal project to address their hypotheses that lesions associated with maternal vascular malperfusion (MVM) recur in subsequent pregnancies and that this pathology explains the majority of recurrent adverse outcomes (RAO). The methodology is sound and the conclusions largely support previous data that (a) MVM has a significant recurrence risk and (b) MVM is not the only placental pathology associated with RAO, With regard to the latter point, it is well known that other pathologies such as VUE, maternal floor infarction, and chronic histiocytic intervillositis are also associated with RAO and that many RAO do not show any consistent pattern of histopathology. Nevertheless, the data presented with appropriate modifications as detailed below provide a fine grained account of the relationships between different types of RAO, MVM, and decreased placental weight that expand our understanding of this important topic. My more specific comments follow:

1. The MVM broad category is fundamentally flawed. The narrow category combining histologic MVM with decreased placental weight is justified, but there are many other placental histologies associated with decreased placental weight (e.g. VUE, maternal floor infarction, and severe examples of fetal vascular malperfusion and chronic abruption) just as there are many small placentas without any specific histologic changes. I suggest that the MVM broad category be split into two subgroups (1) decreased placental weight NOS and (2) placentas showing histologic MVM with weights greater than 10th percentile. If this decreases the power to find an association, an alternative MVM broad group would be any histologic MVM irrespective of placental weight.

2. Abstract line 36: Shouldn't the word "no" be deleted?

3. Line 152: I do not understand how the last two words "for 3865" apply to this sentence.

4. Lines 158-165 and lines 236-242: These sections address the clinical entity of "recurrent pregnancy loss or miscarriage (3 or more losses)", not the less stringent "previous spontaneous abortion" which would be one or more losses.

5. Lines 263-273: It is not clear whether adjusting for MVM used the narrow or broad definition.

6. Line 316: The term "recurrent placental pathology" is inaccurate. Should be "recurrent MVM" since there are many other placental pathologies, such as VUE, fetal vascular malperfusion, and others, that cause the same outcomes, can recur, and are associated with small placentas.

7. Lines 326-327: Reference 50 addresses recurrent preeclampsia, not adverse outcomes.

8. Lines 332-333: There are actually many papers documenting specific patterns of placental pathology in women with recurrent adverse outcomes ( "multiple complicated pregnancies").

9. Lines 347-351: In this section I actually believe that the authors are underestimating the quality of the original data. I have discussed at length the histopathologic coding of the Collaborative Project with its main study pathologist and I would argue that very little has changed since that time. Differences would be changes in terminology, as the authors discuss, and probably some increased sensitivity for detecting mild lesions. Specifically, I do not think that distal villous hypoplasia would have been missed at that time.

6. PLOS authors have the option to publish the peer review history of their article (what does this mean?). If published, this will include your full peer review and any attached files.

Reviewer #1: No

---

## [Author Response · Author response to Decision Letter 0]

10 Jan 2020

I have uploaded a "Response to Reviews" file. I have also pasted the text from that file below.

Dear Dr. Spradley,

Thank you very much for the encouraging response to our manuscript (PONE-D-19-32724), and to the reviewer for their constructive comments. Below, we have outlined our responses to each of the comments, with the reviewer’s comments in italics. 

Sincerely,

Julian

Journal requirements

We have revised the title page to be consistent with PLOS ONE style. We have also revised the names of the supplementary files, and how they are cited within the manuscript.

2. Please note that all PLOS journals ask authors to adhere to our policies for sharing of data and materials: https://journals.plos.org/plosone/s/data-availability. According to PLOS ONE’s Data Availability policy, we require that the minimal dataset underlying results reported in the submission must be made immediately and freely available at the time of publication. As such, please remove any instances of 'unpublished data' or 'data not shown' in your manuscript and replace these with either the relevant data (in the form of additional figures, tables or descriptive text, as appropriate), a citation to where the data can be found, or remove altogether any statements supported by data not presented in the manuscript.

We removed 3 statements with “results not shown”. All three cases stated that similar results were obtained when analyses included institution as a covariate. Since institution does not appear to be an important covariate, we have not included these results. 

3. We noticed you have some minor occurrence(s) of overlapping text with the following previous publication(s), which needs to be addressed:

https://doi.org/10.1016/j.placenta.2019.01.012

https://doi.org/10.1177%2F1093526619852871

In your revision ensure you cite all your sources (including your own works), and quote or rephrase any duplicated text outside the Methods section. Further consideration is dependent on these concerns being addressed.

The first publication listed was authored by one of us (JKC) and describes the same dataset, and so there is some overlapping text in the Methods section. In the revised version, we have cited this previous publication at the beginning of the duplicated text.

We have removed the text associated with the second publication. However, we note that in the case the second publication, the overlapping text was due to the description of a number of specific diagnoses and lesions which could not be rephrased, i.e., the repetition of text was not inappropriate. We have confirmed this with Sarah Mills in the Editorial Office.

Review Comments to the Author

Reviewer #1

1. The MVM broad category is fundamentally flawed. The narrow category combining histologic MVM with decreased placental weight is justified, but there are many other placental histologies associated with decreased placental weight (e.g. VUE, maternal floor infarction, and severe examples of fetal vascular malperfusion and chronic abruption) just as there are many small placentas without any specific histologic changes. I suggest that the MVM broad category be split into two subgroups (1) decreased placental weight NOS and (2) placentas showing histologic MVM with weights greater than 10th percentile. If this decreases the power to find an association, an alternative MVM broad group would be any histologic MVM irrespective of placental weight.

We agree with the reviewer’s concerns regarding the broad category. We originally included this category because it was used by a previous publication, and we wanted to allow comparison of different approaches. However, upon reviewing the literature, we can only find one publication using the “broad” approach defined in this way. We have therefore replaced this category with the “alternate” group proposed by the reviewer (i.e., any MVM histology, regardless of placental weight), which is also an approach that has been used previously by others (as cited in revised version). We have not used the subgroups suggested, as the purpose of these analyses is not to compare mutually exclusive groups of pregnancies with each other (low placental weight with no MVM lesion vs. low placental weight with MVM lesion vs. normal placental weight with MVM lesion), but rather to compare different approaches to categorizing MVM lesions that have been used in the literature.

Reanalysis using the new MVM broad category (any MVM lesion, irrespective of placental weight) yields very similar results to our previous MVM broad category, although the prevalence of MVM lesions is somewhat lower (because small placentas with no histological lesions are no longer considered to show MVM lesions).

2. Abstract line 36: Shouldn't the word "no" be deleted?

It has been deleted, thanks.

3. Line 152: I do not understand how the last two words "for 3865" apply to this sentence.

Text has been reworded.

4. Lines 158-165 and lines 236-242: These sections address the clinical entity of "recurrent pregnancy loss or miscarriage (3 or more losses)", not the less stringent "previous spontaneous abortion" which would be one or more losses.

We have used the term “previous spontaneous abortions” rather than “recurrent pregnancy loss” because the latter term is often defined as 3 or more consecutive losses. We only had information on total losses, not consecutive losses.

5. Lines 263-273: It is not clear whether adjusting for MVM used the narrow or broad definition.

This has been clarified.

6. Line 316: The term "recurrent placental pathology" is inaccurate. Should be "recurrent MVM" since there are many other placental pathologies, such as VUE, fetal vascular malperfusion, and others, that cause the same outcomes, can recur, and are associated with small placentas.

Text revised.

7. Lines 326-327: Reference 50 addresses recurrent preeclampsia, not adverse outcomes.

This sentence has been removed (please see response to following comments).

8. Lines 332-333: There are actually many papers documenting specific patterns of placental pathology in women with recurrent adverse outcomes ( "multiple complicated pregnancies").

We agree, and have removed this paragraph. Upon reflection, the references that we cite earlier in the Discussion provide better context for our study than case studies.

9. Lines 347-351: In this section I actually believe that the authors are underestimating the quality of the original data. I have discussed at length the histopathologic coding of the Collaborative Project with its main study pathologist and I would argue that very little has changed since that time. Differences would be changes in terminology, as the authors discuss, and probably some increased sensitivity for detecting mild lesions. Specifically, I do not think that distal villous hypoplasia would have been missed at that time.

We agree regarding the value and quality of the data, and sought to be conservative in considering the limitations of our study. We have moderated this paragraph in response to the reviewer’s comments. The term “distal villous hypoplasia” is not used in the pathology coding of the dataset, and it is not clear what terms in the NCPP dataset would correspond closely to this lesion, but of course during pathological assessment, it may have been noted and described in other terms.

---

## [Decision Letter · Decision Letter 1]

15 Jan 2020

PONE-D-19-32724R1

Pregnancy complications recur independently of maternal vascular malperfusion lesions

PLOS ONE

Dear Christians,

Thank you for submitting your manuscript to PLOS ONE. After careful consideration, we feel that it has merit but does not fully meet PLOS ONE’s publication criteria as it currently stands. There are still minor comments raised by the reviewer that need to be addressed. Therefore, we invite you to submit a revised version of the manuscript that addresses the points raised during the review process.

We would appreciate receiving your revised manuscript by Feb 29 2020 11:59PM. To enhance the reproducibility of your results, we recommend that if applicable you deposit your laboratory protocols in protocols.io, where a protocol can be assigned its own identifier (DOI) such that it can be cited independently in the future. For instructions see: http://journals.plos.org/plosone/s/submission-guidelines#loc-laboratory-protocols

We look forward to receiving your revised manuscript.

Kind regards,

Frank T. Spradley

Academic Editor

PLOS ONE

Reviewers' comments:

Reviewer's Responses to Questions

**Comments to the Author**

1. If the authors have adequately addressed your comments raised in a previous round of review and you feel that this manuscript is now acceptable for publication, you may indicate that here to bypass the “Comments to the Author” section, enter your conflict of interest statement in the “Confidential to Editor” section, and submit your "Accept" recommendation.

Reviewer #1: (No Response)

2. Is the manuscript technically sound, and do the data support the conclusions?

Reviewer #1: Yes

3. Has the statistical analysis been performed appropriately and rigorously? 

Reviewer #1: Yes

4. Have the authors made all data underlying the findings in their manuscript fully available?

Reviewer #1: Yes

5. Is the manuscript presented in an intelligible fashion and written in standard English?

Reviewer #1: Yes

6. Review Comments to the Author

Reviewer #1: The authors have substantially improved the manuscript and demonstate that MVM, as defined by 1960's criteria, is not the predominant reason for recurrent adverse outcomes in subjects with MVM in an index pregnancy. Nevertheless, I am not sure that it should be concluded that MVM has no significant recurrence risk. I would make a few points:

1. Why is MVM recurrence risk adjusted for gestational age (GA)? MVM can occur at all gestational ages and there is no a priori reason to assume that GA would be a confounder.

2. Odds ratios are consistently elevated for all comparisons of MVM recurrence in Table 2, adjusted and unadjusted. Recurrence is significant before adjustment for GA in the MVM broad category and borderline significant even after adjustment. The lack of significance for the MVM narrow category might be attributed to small numbers (only 2 subjects with recurrence).

3. There is no a priori reason to restrict recurrence to the previous MVM category. The modifiers are continuous not categorical. In other words, recurrence for MVM narrow should include both broad and narrow MVM and recurrence for MVM broad should include both broad and narrow MVM.

I would ask if the authors might accept that MVM has a recurrence rate, albeit small compared to other factors in explaining subsequent adverse outcomes.

7. PLOS authors have the option to publish the peer review history of their article (what does this mean?). If published, this will include your full peer review and any attached files.

Reviewer #1: No

---

## [Author Response · Author response to Decision Letter 1]

16 Jan 2020

The comments below are the same as those in the Response to Reviews file that has been uploaded.

Dear Dr. Spradley,

Thanks very much to you and to the reviewer for your prompt response regarding our revised manuscript (PONE-D-19-32724). Below, we have outlined our responses to each of the comments, with the reviewer’s comments in italics. 

Sincerely,

Julian

Reviewers' comments:

Reviewer #1: The authors have substantially improved the manuscript and demonstate that MVM, as defined by 1960's criteria, is not the predominant reason for recurrent adverse outcomes in subjects with MVM in an index pregnancy. Nevertheless, I am not sure that it should be concluded that MVM has no significant recurrence risk. 

We have modified this conclusion (described further below).

1. Why is MVM recurrence risk adjusted for gestational age (GA)? MVM can occur at all gestational ages and there is no a priori reason to assume that GA would be a confounder.

There are two reasons why gestational age was included in the analysis of recurrence. Firstly, we used the same set of covariates for all analyses for consistency and objectivity, i.e., to avoid analysing the data in different ways and then having to make subjective decisions regarding which analysis to use for each outcome. In the case of MVM, we thought that gestational age might influence the probability of MVM diagnosis (e.g., if some lesions were more apparent later in pregnancy, or if some lesions only develop later in pregnancy). As it turns out, gestational age is a highly significant covariate. 

2. Odds ratios are consistently elevated for all comparisons of MVM recurrence in Table 2, adjusted and unadjusted. Recurrence is significant before adjustment for GA in the MVM broad category and borderline significant even after adjustment. The lack of significance for the MVM narrow category might be attributed to small numbers (only 2 subjects with recurrence).

We have revised the text to acknowledge this (in some cases reverting to text in the original manuscript).

3. There is no a priori reason to restrict recurrence to the previous MVM category. The modifiers are continuous not categorical. In other words, recurrence for MVM narrow should include both broad and narrow MVM and recurrence for MVM broad should include both broad and narrow MVM.

The two categories (MVM broad and MVM narrow) are two different criteria for assessing the presence/ absence of the same pathological entity, MVM. We performed analyses with both sets of criteria to allow comparison of these two criteria because some studies have used one approach while others have used the other. Thus, if one believes that MVM narrow provides a better assessment of the presence of MVM, then it wouldn’t make sense to consider the MVM narrow pathology to have recurred if there was MVM broad (but not narrow) in the previous pregnancy and MVM narrow in the subsequent pregnancy. The MVM narrow cases are a subset of the MVM broad cases (i.e., if a pregnancy is categorized as having MVM using the narrow criteria, it will also be so categorized with MVM broad). Therefore, the analysis of recurrence of MVM broad lesions does include MVM narrow lesions in the previous pregnancy.

I would ask if the authors might accept that MVM has a recurrence rate, albeit small compared to other factors in explaining subsequent adverse outcomes.

The revisions in response to comment 2 above acknowledge the recurrence of MVM broad lesions.

---

## [Decision Letter · Decision Letter 2]

22 Jan 2020

Pregnancy complications recur independently of maternal vascular malperfusion lesions

PONE-D-19-32724R2

Dear Dr. Christians,

We are pleased to inform you that your manuscript has been judged scientifically suitable for publication and will be formally accepted for publication once it complies with all outstanding technical requirements.

With kind regards,

Frank T. Spradley

Academic Editor

PLOS ONE

Reviewers' comments:

Reviewer's Responses to Questions

**Comments to the Author**

1. If the authors have adequately addressed your comments raised in a previous round of review and you feel that this manuscript is now acceptable for publication, you may indicate that here to bypass the “Comments to the Author” section, enter your conflict of interest statement in the “Confidential to Editor” section, and submit your "Accept" recommendation.

Reviewer #1: All comments have been addressed

2. Is the manuscript technically sound, and do the data support the conclusions?

Reviewer #1: Yes

3. Has the statistical analysis been performed appropriately and rigorously? 

Reviewer #1: Yes

4. Have the authors made all data underlying the findings in their manuscript fully available?

Reviewer #1: Yes

5. Is the manuscript presented in an intelligible fashion and written in standard English?

Reviewer #1: Yes

6. Review Comments to the Author

Reviewer #1: (No Response)

7. PLOS authors have the option to publish the peer review history of their article (what does this mean?). If published, this will include your full peer review and any attached files.

Reviewer #1: No

---

## [Editor Report · Acceptance letter]

27 Jan 2020

PONE-D-19-32724R2 

Pregnancy complications recur independently of maternal vascular malperfusion lesions 

Dear Dr. Christians:

I am pleased to inform you that your manuscript has been deemed suitable for publication in PLOS ONE. Congratulations! Your manuscript is now with our production department. 

With kind regards,

on behalf of

Dr. Frank T. Spradley 

Academic Editor

PLOS ONE